# Position: Every Ground Truth is a Human Construction, not an Objective Truth

**Charlotte Högberg** [1]   **Ericka Johnson** [2]   **Kiri L. Wagstaff** [3]

## Abstract

Ground truth datasets play a fundamental role as reference values in the training and evaluation of machine learning models. This position paper argues that ground truths are not neutral objective measurements that are naturally given, but instead that they are constructed by arrangements of humans and technologies. We argue that the ML community will benefit from articulating and discussing these often invisible or unreported choices and acknowledging that reference data sets are contingent, not universal. Focusing on the situated and context-dependent nature of ground truths can improve reliability by enabling a better informed perspective on where, when, and how the datasets, and the models they have shaped, can best be used. We argue for increasing 'situated reliability' which includes articulating the limits and strengths of models and their truth claims. Finally, paying more attention to the construction of ground truths can support transparency, accountability, and interdisciplinary work.

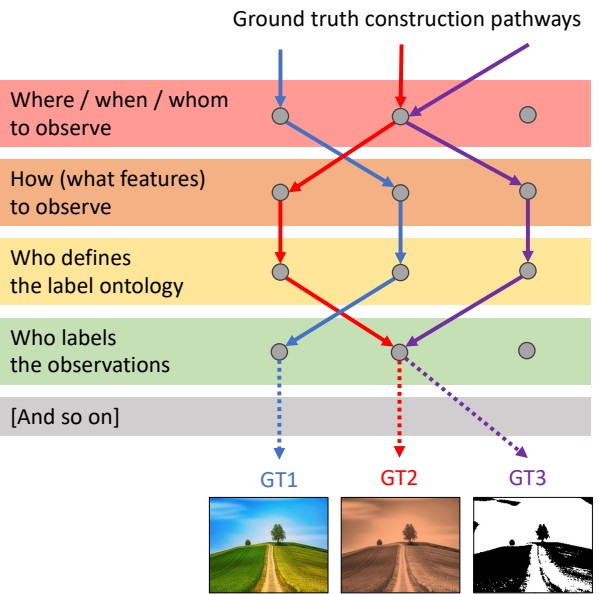

*Figure 1.* Every ground truth is **constructed** as the result of multiple decisions, not a single objective truth. Different pathways yield different ground truths (GT1, GT2, GT3, ...) for the same task.

## 1. Introduction

In machine learning (ML) research and development, the term "ground truth" commonly refers to datasets viewed as containing the true values of a given concept (Kang, 2023) that are used to train and evaluate ML models. An often overlooked aspect is that these ground truths are not neutral objective measurements naturally given. All of our "truths" are constructed. We use ground truth as a countable noun in this paper to emphasize that *a ground truth* for an ML project is only one of many potential *ground truths*. Constructing a ground truth encompasses work beyond just aligning with domain knowledge, yet these additional choices and their consequences are often not discussed or reported.

[1]Lund University, Lund, Sweden [2]Linköping University, Linköping, Sweden [3]Oregon State University, Corvallis, OR, USA. Correspondence to: Charlotte Högberg <charlotte.hogberg@lth.lu.se>.

*Proceedings of the 43rd International Conference on Machine Learning*, Seoul, South Korea. PMLR 306, 2026. Copyright 2026 by the author(s).

**Our position is that ground truths are constructed, and that ML researchers should articulate the choices involved in their construction and discuss the impact those choices have on their results.**

The concept of ground truth has been defined in various ways. The term has been used in remote sensing since the 1960s and been adopted across fields such as computer science and engineering, psychology, neuroscience, and economics (Woodhouse, 2021). One view regards ground truth as the fundamental, real, or underlying facts collected at the source. In remote sensing, it has been defined as "information obtained by direct measurement at ground level, rather than by interpretation of remotely obtained data" (Woodhouse, 2021). A more inclusive definition is "information obtained by direct observation of a real system, as opposed to a model or simulation; a set of data that is considered to be accurate and reliable, and is used to calibrate a model, algorithm, procedure" (Woodhouse, 2021). Kang (2023) argues that the concept basically refers to information assumed to be true in the development of ML models.

We follow Jaton's (2017) definition of ground truths as the result of ground truthing, which is the practice of defining the problem to be solved and what the input values and desired output targets should be (Jaton, 2021; 2017). In this paper, we point out that ground truths are the result of multiple decisions about what sources of knowledge are legitimate, how to interpret observations, what values are recorded, and who makes and validates those attributions (illustrated in Figure 1). All of these decisions and contextual practices impact which ground truths are accepted. Therefore, "ground truth" is a black-box term that comprises many component decisions on inclusions and exclusions of people, technologies, and concepts.

Opening up that black box means recognizing the contingency—the 'it could have been otherwise' aspect—of ground truths. It encourages us to think about the possibility of alternative ground truths. When we see ground truths as plural and produced, we can consider how the legitimacy (or at the very least, the broad applicability) of ML results from ground truths could be undermined due to their nature as contextual, contingent, and situated.

Our starting point is that ground truths are not preexisting entities but are constructed to fit the task addressed by the ML model (Jaton, 2017). They are not solely technical, machine-made, digital objects, or natural facts. Instead, much like the data produced through professional vision in other domains (Goodwin, 1994), they are human constructions (Jaton, 2024) formed by assumptions, organizations, processes, and the use of technology. Flattening the real world into ground truth datasets requires a series of choices and practices involving problem formulation, equipment, settings, expertise, conditions, and epistemologies. It is a version of knowledge making through coding, highlighting, and producing material representation (Goodwin, 1994). This also applies when using previously collected "real-world" data such as medical records, in terms of both their initial creation in the healthcare system and the choices of which data, variables, and labels to include.

By unboxing the black boxes of ground truths, we want to highlight the challenging and necessary work of creating ground truths for machine learning. We urge the machine learning community to stop framing ground truths as neutral, objective, and pre-existing values. They are made through diverse practices that commonly require interdisciplinary communication and shared understanding, in joint creation and partnership. In addition, we stress the need to unpack the notion of the expert-labeler, and we compare views of them as oracle versus as sensor as well as important aspects of engaging the crowd as a label source.

We encourage acknowledgment of and reflection on how ground truths are (necessarily) intentional creations. These practices will increase ground truth quality and thereby the reliability of ML models by (1) Increasing the situatedness and hence the sharpness of a ground truth that is produced in collaborative and conscientious practices and (2) Allowing us to question the limits of applicability for a given ground truth for ML model training. **We call for thinking of ground truth as situated, specific, contingent, and contextual—which will make it better, sharper, more useful—but will also narrow its usability to particular contexts. This perspective can also reduce the uncritical adoption of data sets for inappropriate use cases.** Recognizing the work of ground truthing could also benefit interdisciplinary collaboration and improve evaluation, transparency, and technology adoption.

While some aspects of how benchmark datasets are made and (mis)used have been discussed previously, the current level of awareness around ground truth construction remains insufficient. It has not yet led to agreement on how to characterize and communicate the relevant aspects of ground truth data that are context-specific, situated, and hence could change our conclusions if different data construction decisions were made. There are still no standard vocabulary or documentation methods to express those limitations, nor standard practices for comparing the impact when different choices are made, such as about who serves as the source of labels. Our hope is that this paper, through its recommended actions, dialogue, and researcher training that views "data collection" as "data construction", will strengthen the community's understanding of datasets and their (in)valid uses.

In this paper, we first discuss why this topic matters in Section 2 and review related work (Section 3), then describe different approaches to the construction of ground truths (via the concept of "construction sites") in Section 4. There we analyze the construction of ground truths for different types of reference values, such as direct measurements, expert interpretations, crowd-sourced consensus, and synthetic data. In Section 5, we address aspects of communication and collaboration in constructing sense in ground truthing. We conclude with calls to action (Section 6) and alternative views (Section 7), followed by a summary of our conclusions (Section 8).

## 2. Why Ground Truth Clarity Matters

We argue for the need to acknowledge that ground truths are constructed, and that the process is (by necessity) creative, interpretative, contingent, and therefore subjective. Ground truthing is not the passive capture and representation of neutral, objective facts, and datasets and labels are not mirrors of the natural world. The decisions, choices, assumptions, conditions, and other factors that impact the construction of a ground truth need to be made visible. Ground truth contingency also needs to be openly discussed and reflected upon,

because every ground truth could have been constructed differently and resulted in very different outcomes.

This is important due to the impact that ground truths have on the development, evaluation, and functioning of ML models. Ground truthing is, after all, "a task-bounding process and a form of intentional biasing that hardlines the limits of the algorithm and the possible range of outcomes for an ML system" (Kang, 2023, p. 3). It poses a plethora of limits on both how we understand the world and what ML models do in the world. Ground truth choices made for benchmark data sets can have enormous impact that echoes for decades. For example, the digit images in the benchmark MNIST data set (LeCun et al., 1994) were reduced from 128x128 pixels to 28x28 pixels in the 1990s due to resource limits of computers at the time (LeCun et al., 1998). This has led to the training of thousands of digit classifiers that operate only on 28x28 pixel images. These classifiers are incapable of classifying the original full-resolution images from 1992, much less images from today produced by higher-resolution scanners, despite modern computing environments with orders of magnitude more memory and computational resources. Newly acquired digit images today have to be reduced to this artificially tiny size to suit MNIST-based classifiers, due to a 1990s choice of how to construct a digit ground truth.

**What are the consequences of not acknowledging that ground truths are constructed and influenced by multiple factors?** Ignorance about context can lead to inappropriate or "off-label" (Shimron et al., 2022) use of a ground truth with severe negative, and even harmful, impacts. For example, undocumented data processing steps used to create open-access MRI ground truths create an artificially easier problem, leading to an unfortunate performance drop of 47% when the model was applied to newly collected MRI data (Shimron et al., 2022). Ambiguous annotation processes in dermatology could result in overestimating accuracy and putting patients at risk (Stutz et al., 2025).

In addition, blind use of a pre-existing ground truth may not result in new knowledge, but merely reproduce current practices and domain expertise (Henriksen & Bechmann, 2020). There is also the conundrum of using human-generated ground truths for training and evaluation when aiming to surpass that same human capability (Henriksen & Bechmann, 2020; Högberg, 2025). Moreover, the presumed universality of ground truths (that they are valid across different contexts and domains) often does not hold (Lee & Ribes, 2025). For instance, the well-known UCI Adult dataset was originally collected to demonstrate a dataset visualizer, but has been applied far beyond that, showing the context drift of benchmark datasets (Kohavi, 2025).

We recognize that ground truthing is a necessary, pragmatic step to enable ML development and evaluation (Kang, 2023), manifesting as the "assumptions we need to make"

(Högberg, 2025), and that it is often performed with some understanding of its incompleteness. Yet the ML community needs to openly discuss ground truths as human constructions rather than simply as *the true knowledge*. As Kang (2023, p. 1) argues, "With the increasing complexity of the tasks to which ML has been applied over the past six decades, [...] agreement on what constitutes adequately stable ground truths for ML systems has become exponentially more complicated." In fact, we should discuss whether we should keep calling it "ground truth" at all (Woodhouse, 2021).

Finally, the position that we argue for is highly important for reliability. In line with acknowledging ground truths as constructed, we propose adopting an understanding that they result in a reliability that is **situated** (reliable only in a given context). Situated reliability is not a quantitative value, but instead a key concept that helps us move from discussing reliability as a generic property to understanding it as necessarily "situated" in a particular context. If we continue to employ singular ground truths, we should improve the awareness of the limits and contingencies of truth claims and the ways by which we understand the basis of, for example, measuring accuracy against a certain ground truth. Recht et al. (2019) studied this aspect of the commonly used ImageNet dataset, finding that "changes in the sampling strategy can indeed affect model accuracies by a large amount, even if the data source and other parts of the dataset creation process stay the same." Careful evaluation and documentation practices can make the situatedness of reliability more visible. We need to recognize **situated reliability** as a way to capture the limits of ground truths, whether created by the context of data collection, the expert judgments, the range of sensors, or all other factors shaping its construction.

## 3. Related Work

The construction of ground truths means that their perceived quality and value are subject to negotiations between machine learning practitioners and their collaborators about how to establish the best ground truth for the task. An examination of AI experts' work of establishing ground truths for medical AI (Högberg, 2025) demonstrated the value of using "ground truthing" as a verb (Jaton, 2017; Henriksen & Bechmann, 2020) to describe the joint work of medical and ML expertise. Muller et al. (2021) described the *social* process of generating labeled data as the result of human negotiation. This view is also supported by early lessons learned from studies of the collaborative practices used to create computational tools for real world applications (Collins, 1990; Suchman et al., 1999).

Takeaways from the field of Computer Supportive Collaborative Work include the value of using ethnographic and

ethnomethodological methods (Goodwin, 2000; Raeithel, 1996) when collaborating around technology development to examine how agreement is achieved (Hughes, 1988; Lave & Wenger, 1999), which knowledge artifacts are included or excluded (Star, 1991), and whose framings of a domain are engaged (Orr, 1996; Resnick et al., 1991). This work set the stage for later studies of how computer scientists could work together with domain experts to develop technologies with real world applications (Suchman, 2007; Milligan et al., 2011) and through innovative design methodologies (Costanza-Chock, 2020). These methodologies can inspire analysis and change in ML practices, such as ground truthing, that are proving integral to shaping the development of AI tools now.

Additional aspects of the contingent nature of ground truths include observations that "raw data" is an oxymoron, because data is always "cooked" by processes of collection, use, and analysis (Gitelman, 2013); that data is situated and local (Loukissas, 2019); and that ground truths are not pre-existing entities but shaped to fit the task of the ML model, as in the case of digital image processing (Jaton, 2017). Activities and negotiations of establishing ground truths for ML in healthcare have been conceptualized as truth practices, described as a multi-modal and multilevel performance of truth involving engineers and medical experts with different methods and knowledge (Henriksen & Bechmann, 2020). In such domains, ground truths are assembled and valued based on not only medical expert knowledge, but also temporal and technical qualities, the ability to support generalization, and humanness, as expert-based labels instill both trust and the risk of misjudgments (Högberg, 2025).

Empirical studies have found that a multitude of factors play into the annotation of data to establish a ground truth schema for medical AI, including external (regulations, context of creation and use, commercial and operational demands) and internal factors (epistemic differences and limits of the labeling process) (Zajac et al., 2023). These factors shape and constrain the design of a ground truth schema, and they impact the possibility of achieving responsible AI. Lebovitz et al. (2021) argue for the benefit of deconstructing ground truths to better understand why an given AI implementation is not successful. In a medical setting, they found a critical limitation that they characterized as the ground truth containing the know-what, but not the know-how, of clinical practice. Previous work also addressed how different types of digital reference objects, simulations, and synthetic data are used as ground truths, and in some cases claimed as the 'perfect' or 'known truth', since they can be controlled and offer flexibility (Högberg & Winter, 2026).

Additionally, the reuse of ground truth data sets and algorithms in new cases and domains distinct from where they were generated is common practice, and not unique

to ML (Lee et al., 2025), but could nonetheless be problematic (Lee & Ribes, 2025) with regard to the assumed universality of the constituent ground truths. Awareness of the risk of mismatches between the context of data collection and that of model application has led researchers to advocate for improved data documentation practices, informing about provenance, creation, and use of ML datasets to mitigate harmful outcomes (Gebru et al., 2021).

For large language models, research has shown that user-informed model alignment (reinforcement learning from human feedback or RLHF) is strongly influenced by choices about whose preferences, or value judgments, inform the alignment (Ouyang et al., 2022). The authors warned that the 40 contractors they employed to align a GPT-3 model were "clearly not representative of the full spectrum of people who will use and be affected by our deployed models." This suggests a need for increased discussion of how ground truth construction can shape and limit model performance and generalizability.

## 4. Ground Truth Construction Sites

Depending on the task that the ML model is intended to perform, different levels of expertise are required. For specialized or high-stakes tasks, a high level of expertise becomes necessary to provide reference values with a sufficient level of reliability, such as identifying a malignant tumor on a medical image. For other tasks, common knowledge or non-specialist lay knowledge may be enough, providing the opportunity to construct ground truths by means such as crowd sourcing.

We present and discuss a set of common "construction sites" where ML ground truths are created: direct measurement of a natural phenomenon, expert interpretation of cases or observations, consensus from a crowd, and synthetic data or simulations.

### 4.1. Direct measurement of a natural phenomenon

Collecting data in the form of direct measurements, such as by sensors or instruments (telescopes, microscopes, microphones, cameras, etc.) is commonly seen as the most direct way of obtaining a ground truth. This type of ground truth construction is largely seen as a factual registration of natural states, and less subjective than settings in which labels hinge on human interpretation.

Yet even this collection of "true" values is dependent on an array of choices and preconditions. These include necessary decisions such as: what equipment to use (resolution, sensitivity, data rate, etc.), where to place sensors, when and how often to perform the measurement, how values should be collected and what annotations and metadata to include (what the set of appropriate labels is), and how generalizable the

measurements and labels are. These decisions may also be influenced by factors unrelated to the phenomenon studied, such as the availability of tools, computational resources, and/or funding. Different choices will result in different ground truths for the same problem.

An example demonstrating the contingent nature of ground truth appears in a study that sought to catalog all fresh impact craters on Mars (Daubar et al., 2022). "Fresh" impact craters were defined as those whose creation time could be bounded, i.e., those craters that had both "before" (no crater) and "after" (with crater) images in the data set. As a result, it is likely that some genuinely fresh impacts were labeled as "not fresh" due to the temporal limits of the data set, independent of the appearance of the crater in a given image. It is evident that a different source data set, or a different definition of "fresh" impact craters, would lead to a different ground truth.

In another case, investigators trained an ML model to classify the Mars surface into 14 terrain types. After finding that this model only achieved 74% accuracy, they created a second ground truth that assigned the same pixels to 5 coarser-grained classes. The new trained classifier achieved 92% accuracy (Barrett et al., 2022). It is not possible to say which ground truth is the "right" one without knowing the context of use. For the motivating case of Mars rover landing site selection, mission planners would need to decide what level of accuracy they need to trust a classifier's output, as well as which class distinctions are critical for that task (are 5 classes sufficient, or were all 14 needed?).

When not enough data from distant planets are available, analog sites on Earth are sometimes used as substitutes in data collection. This poses additional questions about how measurements of such sites can reliably function as "truth-spots" (Ostrowska, 2026) and training data for ML.

### 4.2. Expert interpretation of a case or observation

Another common way to construct ground truths is by collecting post-hoc expert interpretations that are treated as true labels, especially for tasks requiring highly-skilled domain expertise. However, expert-labeled data may be employed later by investigators who lack the same expertise, especially for benchmark datasets. One example is the commonly used multivariate, categorical benchmark dataset of physical characteristics of mushrooms that are classified as edible or poisonous (Schlimmer, 1981). Labels were derived from expert knowledge in the Audubon Society Field Guide to North American Mushrooms. Mushrooms with "unknown edibility" were conservatively assigned to the poisonous class. The binary labels of "edible" versus "poisonous" have been treated as a ground truth ever since, and knowledge of which ones were of unknown edibility has not been preserved. Could this choice have introduced

errors? The situatedness (what expertise was available at that time and place) and representation choices (number of classes) that shaped this ground truth matter. We do not know how many ML papers using this dataset would have reported different conclusions had all three original classes been used.

In the field of medicine, ML is often applied to separate the healthy from the pathological. For image-based data, it often involves the task of detecting signs of disease. Medical or clinical expert interpretations are commonly used to establish ground truths. This is a laborious process when datasets are produced from scratch. In one case, experienced neurologists were recruited to spend hours painting areas of imaged slices of a brain to construct a ground truth for developing brain segmentation algorithms (Högberg, 2025). The products of this expert labor (hand-segmented images) were subsequently treated as the true values.

Another example is the task of identifying malignant tumors in medical images, where image data are labeled as healthy versus not healthy by multiple radiologists. However, this type of collection of accurate labels opens up different types of uncertainty, such as that of inter-expert variability, where radiologists disagree, or intra-expert variability, where the same radiologist gives different assessments in repeated interpretations. In addition, not all expert sources are the same: elite hospitals may be perceived as generating the preferred (best) ground truth (Högberg, 2025).

When expert knowledge is required, the human expert labeler is generally assumed to be (1) an absolutely reliable source of knowledge (oracle) and (2) an impersonal neutral sensor that registers, detects, and provides data in an objective manner. Yet the processes and issues involved in getting reliable labels from human experts are never free from subjectivity. Muller et al. (2021)) observed that in practice, the names and meanings of labels are not fixed, but "malleable, changeable, and negotiable depending on who is applying them." As such, they also become more than simple definitions. Rather, the definitions and the data they are applied to are subject to pragmatic limitations, persons, and organizations. This makes visible the work required to describe the world. Muller et al. (2021) argue that ground truth looks less like an objective truth, and more like the output of a social process. Their analysis shows that the social complexity (and more or less open negotiations) is sometimes necessary to come to agreement about what the data conveys.

Finally, expert-based medical ground truths contain human uncertainties that can introduce errors or inconsistencies, such as in radiology (Lebovitz et al., 2021). In a similar vein, Jaton (2023) describes how the making of a benchmark dataset for personalized cancer immunotherapy resulted in a ground truth of somewhat contested accuracy, yet it re-

mained in use due to it being the only functional benchmark for the task. ML practitioners and researchers using data and labels from existing collections of expert judgments as ground truths, such as open datasets or medical register data, are impacted by these inherent uncertainties and choices about how the data was documented and what to include.

### 4.3. Consensus from a crowd

Ground truths are also constructed by engaging crowds to provide post-hoc labels and confirm, or vote on the accuracy of, the assigned labels (Cabitza et al., 2023). Crowds can be recruited as annotators for tasks requiring lay knowledge or some types of expertise. There are citizen science initiatives such as BirdNet, which invites birdwatchers to contribute sightings, sounds, and labels (Sullivan et al., 2009). However, much work is needed for crowd-sourcing to generate reliable ground truths (Rhee, 2025; Denton et al., 2021).

Some large advances in AI have been achieved by recruiting paid annotators through crowdsourcing marketplaces, such as Amazon Mechanical Turk. One example is the creation of the ImageNet dataset (Deng et al., 2009), which relied heavily on outsourcing the annotation of thousands of images to humans who were instructed to confirm the presence or absence of a given concept in an image. In the final dataset, images were included if sufficient agreement was obtained among the annotators (Denton et al., 2021). The image annotation effort was distributed across ∼49,000 workers from 167 countries. This arguably very culturally diverse pool of workers were tasked with making meaning from data. The assumption was that their annotations would reflect universal understandings and interpretations of the data, despite the wide variety of backgrounds they brought to the task (Denton et al., 2021).

Deriving a single ground truth from a crowd is challenging. For example, in Natural Language Processing (NLP) tasks, the meaning of words, sentences, and statements can be interpreted very differently by different annotators and be highly subjective and cultural. For sentiment analysis, the judgment of whether a statement is positive, negative, or neutral could vary significantly between labelers. Plank (2022) highlights this as human label variation with regards to NLP specifically but also stresses it as a general matter for ML and Computer Vision (CV) and for all stages of the pipeline: data, modeling, and evaluation.

Human label variation (HLV) could be caused by inattention or insufficient expertise leading to errors, but some differences cannot be dismissed as mistakes. HLV can also arise from differences of opinion, concept ambiguity, subjectivity, multiple correct options, or the fact that cultural understandings and responses to an object or emotion shift across time, space, and context (Rhee, 2025). Commonly, variation is "resolved" by aggregation into a majority vote,

allowing only for one belief, label, or category, obfuscating the real-world complexity (Plank, 2022; Cabitza et al., 2023). Plank (2022) identifies the limitations of optimizing and evaluating by a single ground truth, suggesting it might hamper progress, and argues for preserving and using the variation as informative instead of seeing it as a problem, noise, or disagreement to be removed (Plank, 2022; Cabitza et al., 2023).

We currently lack standard ways to document the subtleties and limitations of a ground truth created by crowd-sourced labels. This is especially problematic for data sets that become community-wide benchmarks, employed by thousands of other researchers who had no direct involvement in the ground truth construction process. For example, the ImageNet-1k documentation[1] does not explain enough of the ground truth construction choices (sampling, sources, classes, labelers) to enable others to determine whether it is an appropriate ground truth for their needs. While some more details appear in the accompanying paper (Russakovsky et al., 2015), there is no discussion of the impact of alternative choices and that ImageNet (like any dataset) has reliability that situated in its choice of sources. It is a convenience sample, comprising voluntarily posted images on Flickr and other websites in the 2000s; it is not representative of all human photographers (or image classes, or labelers). Recht et al. (2019) found that different image sampling strategies led to very different ImageNet classifier evaluations. We advocate that this kind of sensitivity analysis be standardized.

### 4.4. Synthetic data, simulations, and "known truths"

Data created by generative AI models and simulations are also utilized as ground truths (e.g., Hamarneh et al., 2008; Högberg & Winter, 2026). Awareness of the generation practices for these synthesized ground truths is necessary to understand the limitations of their use. Synthetic ground truths frequently contain some elements of fabrication, such as data augmentation by generating plausible variations to boost generalizability or data imputation to accommodate missing values.

Other examples include the injection of artificial cancer nodules into real lung scans to evaluate the detection abilities of CNNs and humans (Schultheiss et al., 2021) and the injection of artificial noise in labels to enable controlled studies of model robustness to such noise (de Vries & Thierens, 2025). Synthetic data have also been used to address biased datasets, perhaps most famously for facial identification algorithms (Cascone et al., 2025) and other computer vision use cases, like training autonomous vehicles (Tsirikoglou et al., 2017).

---

[1] https://huggingface.co/datasets/ILSVRC/imagenet-1k

One member of our author team is conducting research on scientists' experiences of using synthetic data. This work finds that two very different modes of ground truthing emerge. The first is to generate a synthetic 'ideal case' of ground truth (Daston & Galison, 2010) using models of scientific phenomena. Example domains where this occurs include protein biology and climate research (Tong et al., 2025). The second approach is to combine synthetic data with original data to obtain a larger, pooled training set. In this process (which often occurs in medical diagnosis domains), the synthetic data are viewed as containing as yet unidentified ground truths in the generated variations. As in Section 4.2, domain experts are then engaged in evaluating and identifying the ground truths of the dataset, which is composed of real and synthetic data.

**These four construction sites show different ways in which ground truths are constructed.** They are each subject to a multitude of choices, actors, equipment, and contingencies that shape the datasets and thereby the ML models they inform. They also raise the necessity of communication and shared understanding between different actors to establish any ground truth dataset.

## 5. Communication and Collaboration in Ground Truth Construction

To construct reliable ground truths requires effort and coordination. This is true whether collecting new sensor data, using data from existing infrastructures, or generating new synthetic data. This work indirectly or directly involves multiple actors, such as the ML practitioners, the data collectors (if different from the ML practitioners themselves), and the labelers and annotators (experts or crowd-sourced actors) assigned to identify the truth in the data. Processes like deciding what the relevant data and sources are, and labeling and annotating data, require communication, agreement, and alignment between different actors, to construct sense of both the data and the concept to be modeled.

### 5.1. Constructing sense

While the verbs 'making' and 'constructing' are often used interchangeably, there is an important difference of nuance. 'Making sense' can indicate that one is creating understandings of an ontologically separate and discrete ground truth: it implies that the ground truth can be seen, identified, and captured by labeling it, requiring the assumption that a ground truth exists independently and the goal is to correctly identify it. We argue that it is more appropriate to refer to 'constructing sense' from the data.

This idea refers to the process of aligning the needs of the ML expert (to create functional categories and labels) and the domain expert (to express an understanding of the world,

derived from historical and contextual knowledge of their field) to *construct* a ground truth, not merely to label it. This involves constructing meaning for the data, the deriving model, and the phenomena being modeled. Changing the terminology from making to constructing acknowledges the situatedness and—to some degree—the arbitrary aspects of ground truthing.

One process in which sense is constructed is the labeling of data. When labeling and annotating data to construct a ground truth, all actors must have a shared understanding of what is being labeled, the goal of labeling, and how to operationalize it. Commonly, this means developing and adopting annotation guidelines and instructions to reach consistency and agreement of which label to use when, what the labels entail, and what annotation should look like (Engdahl, 2024). This is a contingent communicative process to navigate when working with expert labelers (Muller et al., 2021) as well as crowds (Denton et al., 2021). When engaging crowds as labelers, reaching a shared understanding might be even more challenging considering the heterogeneity and distributed nature of the crowd. Constraints on how labeling should be performed also depend on the cost and time restrictions of the ML project, sometimes leading to the use of platforms like Amazon Mechanical Turk (Denton et al., 2021). In these highly distributed cases, often with no face-to-face interaction, good communication practices are vital for constructing reliable ground truths.

With regards to what is treated as ground truth, we urge the field to develop a more reflective and critical review process for data, sources, labels, and documentation for ML development and evaluation. This is especially crucial when the ground truth dataset is collected from existing data infrastructures that were created for purposes other than ML development, such as for census information, biodiversity, or medical record keeping. An extra challenge arises when the original dataset creators are unavailable or uninvolved in the ML project. A critical review is necessary with regards to what the data can convey and how labels can function as reliable ground truths, before being appropriated for ML.

The dataset documentation often functions as a standalone communication tool to establish an understanding of what the dataset includes and what variables and labels mean. We may hope that the documentation is correct and complete while also questioning what has gone undocumented or undiagnosed (Högberg, 2025). Moreover, even when the documentation was written with ML development in mind, there can be practical, organizational aspects of such data collections and labels that prevent them from functioning as the accurate record of facts they are assumed to be when later employed as ground truths. For example, in practice, the same label could have been used somewhat differently due to organizational variations between hospitals.

When possible, it is desirable to engage directly with domain experts. Such collaboration can also inform decisions about the most suitable data, data sources, and labels. This requires highly functioning communication and a shared vocabulary between domain experts and ML practitioners. Often, an ML practitioner alone cannot properly interpret the performance of the ML model with respect to the original domain and must rely on expert knowledge. This requires pairing the goal of the machine learning task with domain expertise about why the task matters (Wagstaff, 2012).

We argue that **more discussion, reflection, and consideration is needed** with regard to **ground truthing as a communicative process of creating shared understandings and constructing sense.**

### 5.2. Joint creation and partnership

The construction of ground truths is dependent on and shaped by *joint* creation and partnership. We argue that the machine learning community should to a greater extent acknowledge the array of actors that contribute to the construction of ground truth.

Establishing a reliable ground truth requires interdisciplinary or intersectoral work. The expertise of machine learning and the problem domain must be combined when formulating the problem to solve, determining what outcomes are useful, choosing data sources, and establishing meaningful categories for algorithmic tasks. This work requires interdisciplinary competencies, including facilitation across knowledge paradigms and insights into the complexities of creating data from the world and knowledge from data. For models to make a positive impact, it is necessary to understand what ground truth is meaningful and useful for the context of model application, something that ML practitioners may not be able to decipher on their own. To facilitate collaborative practices in ground truth construction (and beyond), we suggest that ML education programs to a larger extent include courses or modules in Computer Supported Collaborative Work and fields like Science and Technology Studies and Critical Data Studies.

In addition to the challenges already discussed, ground truthing must comply with ethical and legal restrictions when using ground truths constructed by others (Kroes & Verbeek, 2014; Sobel, 2021). Here, computer science can learn from other fields that have been more confronted with a history of unethical data collection, such as medicine or population science. We suggest that the ML community foster more discussion about the ethical and legal considerations of ground truth constructions and uses. This should be a standard practice and preferably a collaborative task together with domain experts and social science experts (Prainsack & Steindl, 2022; Baumgartner et al., 2023).

Recognizing the elements of joint creation and the collaborative aspect of ground truthing foregrounds a number of insights: It matters **who is in the constellation of actors** constructing ground truths; the **responsibility** for creating ground truths is **distributed**, but not necessarily shared equally, between different members of the constellation; even if an expert or a crowd is attributed with the authority to create 'correct' ground truths, the ML community also shares responsibility for how those truths are made and operationalized.

## 6. Call to Action

Given the issues presented in this paper, we advocate for: (1) greater acknowledgment of ground truths as constructed, (2) formulating multiple ground truths for the same problem and comparing evaluation results on each, (3) developing a standard vocabulary on ground truth limitations, and (4) a standard for how dataset creators should specify relevant contexts for use. These steps can inform (5) improvements to ML courses and educational programs. In more detail, we propose that the ML community should:

1. **Acknowledge the decisions and work involved in producing ground truths** and the fact that they are socially constructed and not just observed or recorded.

2. Recognize that **ground truths are situated** and hence restricted to a **situated reliability**. This can improve knowledge on where, when, and how ground truth datasets, and the models they have shaped, can be of best use. This includes to:

   - Consider how ground truths are **contextual in space and time, reflecting the positionality of people and organizations** making the data, the decisions about categories, the vocabularies and the practical conditions. This necessarily leads to considerations about how frequently ground truths might require updates or revisions.
   - **Embrace humility about the generalizability** of our models to prevent misuse and negative impacts, and **consider the situatedness** of ground truths in **evaluation practices**, which means that the model might need to be evaluated against additional ground truths. This might increase costs in terms of constructing the additional datasets as well as the evaluation runtime. However, these increased costs can be weighed against the costs of inadvertently over-estimating performance and the societal and personal costs of uncritical adoption of ill-fitting ML solutions that could have potentially severe consequences.

3. **Improve our ability to discuss and reflect** upon the construction of ground truths. Because of the situatedness of ground truths, the models that are trained and evaluated on them are also limited. Including additional data in the training set is not always going to solve the model's limitations. Hence:

   - We need **a language** (terminology) to speak about limits of aspects such as the expertise of the model. This can be encouraged by activities such as special issues and workshops dedicated to this topic.

4. Promote a reflective, critical, and open approach to **reporting and documenting** how ground truths were constructed, their limitations, and how these choices may impact the ML results. For example, the Datasheets for datasets' (Gebru et al., 2021) questions about the purpose, collection practices, and limitations for future use can be augmented with questions about additional (necessary) choices in the ground truth construction process. Dataset creators could list the conditions necessary for their ground truth to be properly aligned with a new setting. This can be achieved by:

   - Including **precise information** about datasets and their construction choices in **dataset documentation** and metadata.
   - **Articulating** how other choices in ground truthing could have resulted in different outcomes, and **addressing uncertainties**.
   - **Creating** multiple different ground truths and **evaluating** the impact of those choices on the resulting models, as done by Recht et al. (2019).
   - Adding **critical reflection** on the ground truth to the **list of criteria** for ML paper submissions (checklists) and reviews.

5. **Improve ML courses and educational programs** to include critical reflections on ground truths, training on how to collaborate on knowledge and data construction across domains and paradigms, and legal and ethical issues related to ground truth construction and use.

## 7. Alternative Views

First, one practical/pragmatic objection to our argument is the view that **data sets utilized as ground truths by machine learning are the best representations of natural facts, and not constructed**. However, careful reflection on all of the steps, choices, and actors involved in creating a ground truth reveals that alternatives would lead to a different ground truth. Ignoring the contingency of the construction process limits our ability to conceive of and enact different, possibly better, choices.

Second, when dealing with expert-based ground truths, one can argue for viewing **the expert as an oracle**, always able to deliver accurate predictions under all circumstances, or **the expert as a sensor**, able to perform neutral recording of phenomena and provide accurate measurements. Both views would suggest that ground truths are objective rather than constructed. However, these views do not hold given findings about inter- and intra-observer variability of medical experts' assessments (Soyer, 2018). The ground truths obtained by engaging crowds, with diverse levels of expertise, likewise contain label variation. Hence, that **the crowd can function as an objective sensor** is also an unreliable assumption. Moreover, human label variation is not something that can be fully abolished, and it may be regarded as an opportunity rather than a problem (Plank, 2022).

A third objection to our argument is that **dataset contingency could be solved through more data, or better multimodal data**. Even if this might result in models with increased performance across different contexts, we argue that there is no solution that can eliminate the contingency of ground truths that we have presented.

Finally, some might argue that our recommendations of acknowledging and documenting the construction of ground truths requires **too much work** or is **too much of a burden**. Our response is that it indeed requires some work, but that work is necessary if we want ground truths to be meaningful, reliable, and used appropriately. Recognizing ground truths as constructed, with situated reliability, allows the resulting models to be properly applied in the right contexts.

## 8. Conclusions

We argue for the need to acknowledge and reflect on the fact that ground truths for machine learning are constructed artifacts, in contrast to the common view of them as objective, naturally given truths. An increased focus on the situatedness and context-dependence of ground truths, and the work behind them, can improve the reliability of ML models through a better informed perspective on where, when, and how ground truth datasets, and the models they have shaped, can safely be used. This informs about the limits and strengths of models and their truth claims by defining a 'situated reliability'.

We also raise the question of whether we should think only in terms of singular ground truths. At the very least, we should discuss the work involved in constructing ground truths, what a good ground truth is, and whether there is a need to consider using multiple truths. Finally, paying more attention to the construction of ground truths can offer greater possibilities to achieve transparency of ML models and improve interdisciplinary work in ML development through better communication.

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
