# OpenReview forum: "Position: Every Ground Truth is a Human Construction, not an Objective Truth"
_ICML.cc/2026/Position_Paper_Track — ICML 2026 Position Paper Track regular_

### Official Review · Reviewer_u81f · 2026-03-10

**Significance:** 3
**Argument Clarity:** 3
**Ethics Flag:** Yes
**Rating:** 5
**Confidence:** 4

**Questions:**

How do the authors plans to practically apply 'situated reliability' to actually change model evaluation and paper review, rather than just keeping it as a theoretical concept?"

**Alternative Views Section:**

Yes

**Compliance With Llm Reviewing Policy A Conservative:**

Affirmed.

**Discussion Potential:**

2

**Final Justification:**

Thanks for taking time to answer my questions. I appreciate the author's response by showing some concrete examples, please do put these examples (and maybe elaborate a little bit more if you have space) in the final draft.

Since my original rating is 5 (accept), I'll keep my score.

**Paper Summary:**

This paper argue that "ground truth" in machine learning is never a perfectly neutral fact. Rather, it is constructed by human and technical choice. They propose that model reliability is "situated," meaning label validity depend on its specific context. The paper call for better documentation and interdisciplinary collaboration. The main objective are to improve transparency and ensure safer deployments of machine learning system.

**Position:**

Yes

**Position In Title:**

Yes

**Related Work:**

3

**Strengths And Weaknesses:**

## Strengths
1. The paper is well written.
2. The paper motivate this core concept in a very accessible way.
3. Organizing the discussion around four distinct "construction site" provides an effective framework to understand different data types.

## Weaknesses
1. I think the paper's position is very important and constructive. The only main limitation is the lack of true novelty. Many field (for example, I work on medical AI) already expects rigorous discussion about label definitions and dataset origins in the manuscript. Thus, the paper often read more like a summary of existing concerns to me rather than a completely new perspective. Although this might not be that obvious to the other fields, I still suggest the authors explicitly show (eg, with some concrete example) what current machine learning practice in some fields get wrong.
2. Some proposed solutions sounds a bit abstract. Recommending better reflection is logical, but these is not translated into concrete protocols to report in a manuscript. For instance, the text do not specify what minimum metadata are required or how to handle annotator disagreements. Without an operational framework, the recommendations remains difficult to implement in actual review practice.

**Support:**

3

---

> ### Author Rebuttal · Authors · 2026-03-30
>
> Thank you for the careful review of our paper.
>
> > Q: How do the authors plan to practically apply 'situated reliability' to actually change model evaluation and paper review, rather than just keeping it as a theoretical concept?
>
> We welcome the chance to clarify this concept and make it more concrete. Situated reliability is not a quantitative value, but instead a key concept that helps us move from discussing reliability as a generic property to understanding it as necessarily “situated” in a particular context (see the final paragraph of Section 2).  Evaluation practices that can make the situatedness of reliability more visible include, as mentioned in the paper, evaluating against several datasets, employing a plurality of ground truths rather than upholding an illusion of one possible single ground truth, and measuring uncertainty. We also advocate for better documentation practices to characterize the relevant context (see Calls to Action 3 and 4).
>
> > W1a: The only main limitation is the lack of true novelty. Many field (for example, I work on medical AI) already expects rigorous discussion about label definitions and dataset origins in the manuscript. Thus, the paper often read more like a summary of existing concerns to me rather than a completely new perspective.
>
> We agree that this topic has been raised (see Section 3), yet (vexingly) the current level of awareness remains insufficient, since it has not yet led to agreement on how to characterize and communicate the relevant aspects of “ground truth” data that are context-specific, situated, and (therefore) could change our conclusions if different data construction decisions were made.  Documentation quality is uneven, leading to problematic examples like the Wisconsin breast cancer dataset mentioned in our response to reviewer NiQW.  Ground truths are still mainly treated as singular, as objective facts, and with little or no acknowledgement of the situatedness of its construction and how it impacts reliability (see e.g., Plank (2022) and Muller et al. (2021), cited in the paper). Dataset documentation generally does not address the rows shown in Figure 1.  It is our hope that this paper, through the recommended actions as well as dialogue (and researcher training) that views “data collection” as “data construction”, will strengthen the community’s understanding of datasets and their (in)valid uses.
>
> > W1b: Although this might not be that obvious to the other fields, I still suggest the authors explicitly show (eg, with some concrete example) what current machine learning practice in some fields get wrong.
>
> We included concrete examples of missteps in Section 2 (artificial constraints on modern digit classifiers from MNIST, generalization failures in MRI and dermatology classification tasks) and Section 4 (crater and terrain classification limits, questionable mushroom classifiers, etc.).  See also our discussion of the ImageNet and breast cancer datasets in our response to reviewer NiQW.  We can include these examples in the paper as well.
>
> > W2: Some proposed solutions sounds a bit abstract. Recommending better reflection is logical, but these is not translated into concrete protocols to report in a manuscript. For instance, the text do not specify what minimum metadata are required or how to handle annotator disagreements. Without an operational framework, the recommendations remains difficult to implement in actual review practice.
>
> We too are eager to see specific practices and standards to improve how we construct and document ground truth datasets. What is sufficient minimum metadata will likely depend on the specific context, considering the assumptions that are incorporated into labels and annotations and how they will impact the reliability for the application setting. We see the development of a standard vocabulary (our Call to Action 3) as the necessary first step, which can inform improved documentation methods (Call to Action 4), characterization of situated reliability (Call to Action 2), and better researcher training (Call to Action 5). We can improve the presentation of these steps to convey this envisioned sequence of solutions.

---

> > ### Author Rebuttal · Reviewer_u81f · 2026-04-01
> >
> > Thanks for taking time to answer my questions. I appreciate the author's response by showing some concrete examples, please do put these examples (and maybe elaborate a little bit more if you have space) in the final draft.

---

### Official Review · Reviewer_NiQW · 2026-03-12

**Significance:** 2
**Argument Clarity:** 2
**Rating:** 3
**Confidence:** 3

**Questions:**

Please, see Weaknesses.

**Alternative Views Section:**

Yes

**Compliance With Llm Reviewing Policy A Conservative:**

Affirmed.

**Discussion Potential:**

2

**Final Justification:**

I thank the authors for the answers. The rebuttal discussion clarified and addressed some of my concerns.

I carefully read the replies and communication with other reviewers, but I lean towards maintaining my score.

It feels like a circular problem: the dataset's description should provide "enough" information, but what constitutes "enough" is determined by the practitioner who uses it.
Since the dataset creator could not foresee all use cases, the dataset description will always be "incomplete".
Then, the best one can do is to provide "complete" descriptions, but they will be "lengthy" (as in the ImageNet example) and may still miss important details for practitioners. And this rounds the circle.

**Paper Summary:**

The paper advocates a position that every ground-truth dataset results from specific assumptions and design choices that are typically left implicit. However, these choices could introduce biases and noise, and therefore, training a model based on them could yield unfair, misleading, and potentially even harmful results.

Authors call on the ML community to recognize the issue and be explicit about the assumptions and methodologies when constructing a ground-truth dataset.

**Position:**

Yes

**Position In Title:**

Yes

**Related Work:**

2

**Strengths And Weaknesses:**

## Strengths

The paper raises an important topic and provides good advice and recommendations for practitioners to work carefully with datasets.

## Weaknesses

Although I share the position and some ideas presented in the paper, I believe the text is currently written "too generally", without concrete examples and demonstrations.

I think no one would disagree that it is essential to understand how the datasets we use for training/evaluation are constructed. And no one argues that training a model on biased data would result in a biased (judgment-wise) model. For example, the problem is well identified in the NLP community (see e.g., [1,2]).

The main questions one can ask are: what can one practically do about it? How should we specifically proceed to prevent practitioners from misusing a dataset?

Authors have written several recommendations in the Call to Action section, e.g., "recognize that ground truths are situated", "improve our ability to discuss and reflect upon the construction of ground truths", "promote a reflective, critical, and open approach to reporting and documenting", etc. These are generally good pieces of advice.
But isn't it true that most public datasets used for benchmarking already provide a detailed description of the data-gathering process?

Maybe a concrete example or demonstration from the authors would help me to understand better the problem they are referring to.
In the introduction, the authors say: "Our position is that ground truths are constructed, and that ML researchers should articulate the choices involved in their construction and discuss the impact those choices have on their results". Can the authors demonstrate how they see the ideal way ML research should proceed? Given, e.g., an MNIST dataset, how should ML researchers describe their results?


----

References:

[1] Bolukbasi, T., Chang, K. W., Zou, J. Y., Saligrama, V., & Kalai, A. T. (2016). Man is to computer programmer as woman is to homemaker? debiasing word embeddings. Advances in neural information processing systems, 29.

[2] Kaneko, M., & Bollegala, D. (2022, June). Unmasking the mask–evaluating social biases in masked language models. In Proceedings of the AAAI conference on artificial intelligence (Vol. 36, No. 11, pp. 11954-11962).

**Support:**

2

---

> ### Author Rebuttal · Authors · 2026-03-30
>
> Thank you for your review and insightful comments.
>
> > W1: Although I share the position and some ideas presented in the paper, I believe the text is currently written "too generally", without concrete examples and demonstrations.
>
> You may already be familiar with this problem and its importance. To aid readers with less familiarity, we provided concrete examples of negative impacts in Section 2 (artificial constraints on modern digit classifiers from MNIST, generalization failures in MRI and dermatology classification tasks) and Section 4 (crater and terrain classification limits, questionable mushroom classifiers, subjective judgments from medical experts, cultural variations, etc.).
>
> > W2: I think no one would disagree that it is essential to understand how the datasets we use for training/evaluation are constructed. And no one argues that training a model on biased data would result in a biased (judgment-wise) model. For example, the problem is well identified in the NLP community (see e.g., [1,2]).
>
> It may be that few would conceptually object to the need of understanding how ground truth datasets are constructed, but unfortunately current practices of data documentation, reporting, and claims of generalizability suggest otherwise. Ground truths are still mainly treated as singular, as objective facts, and with little or no acknowledgement of the situatedness of its construction and how it impacts reliability (see e.g., Plank (2022) and Muller et al. (2021), cited in the paper).
>
> > W3: The main questions one can ask are: what can one practically do about it? How should we specifically proceed to prevent practitioners from misusing a dataset?
>
> We advocate for: formulating multiple ground truths for the same problem and comparing evaluation results on each (Call to Action 2), developing a standard vocabulary (Call to Action 3), and a standard for how dataset creators should specify relevant contexts for use (e.g., the rows indicated in Figure 1) (Call to Action 4), for example, the Datasheets for datasets' [Gebru et al., 2021] questions about the purpose, collection practices, and limitations for future use can be augmented with questions about additional (necessary) choices in the “ground truth” construction process. Dataset creators could list the conditions necessary for their "ground truth" to be properly aligned with a new setting - e.g., "this dataset contains only springtime observations, so using it to help analyze other seasons is risky". These steps can inform improvements to ML educational programs (Call to Action 5).  We can include these additional details in the Call to Action section.
>
> > W4: Authors have written several recommendations in the Call to Action section [...]. These are generally good pieces of advice. But isn't it true that most public datasets used for benchmarking already provide a detailed description of the data-gathering process?
>
> This is exactly what we advocate, but it is not the reality yet. Consider two well known datasets:
>
> (1) ImageNet (ILSVRC 2012, https://huggingface.co/datasets/ILSVRC/imagenet-1k): The Kaggle description is quite lengthy yet does not mention who the image creators were, how they were chosen, collection method, how the 1000 classes were chosen, who the labelers were, a discussion about the impact of alternative choices, etc.  While some more details are included in the accompanying paper [Russakovsky et al.,, 2015], the length of it also underscores the lack of standard vocabulary to express dataset context.
>
> (2) Wisconsin breast cancer diagnosis (https://archive.ics.uci.edu/dataset/17/breast+cancer+wisconsin+diagnostic):  A dataset containing measurements of breast cancer tumors labeled “malignant” or “benign”. Documentation describes how features were measured, but not patient demographics, inclusion and exclusion criteria, how samples were labeled (nor does the source paper), so evaluating whether this dataset will generalize to a new set of patients is impossible.
> Note: our position does not apply only to benchmark datasets, but all ground truthing in ML.
>
> > W5: Can the authors demonstrate how they see the ideal way ML research should proceed? Given, e.g., an MNIST dataset, how should ML researchers describe their results?
>
> Yes! The ideal process would include the steps described above for W3. For example, the MNIST dataset should contain enough information for others to decide if those handwriting samples are relevant to a new set of users (and OCR settings), classifiers should be trained on different kinds of digit ground truth to assess the impact of dataset construction choices on performance, and such impacts should be reported in the “Limitations” section of a paper. We expect that these steps will greatly increase reliability and usability of the resulting models.
>
> References:
> Gebru et al., Datasheets for datasets. Commun. ACM 64:12, 86–92, 2021.
> Russakovsky et al., ImageNet Large Scale Visual Recognition Challenge. IJCV, 2015.

---

> > ### Author Rebuttal · Reviewer_NiQW · 2026-04-04
> >
> > I thank the authors for their reply.
> > I am OK with the answers to Weaknesses 1-3.
> >
> > Yet, the answers to Weaknesses 4/5 sound "too general" (as I noted in W1, speaking about the position itself).
> >
> > What I wanted the authors to demonstrate in their answers is the exact implementation of their Call to Action, applied to the description of a dataset. To have a concrete reference.
> >
> > Currently, I find the reply regarding MNIST vague. The authors say "MNIST dataset *should contain enough information* for others to decide...". What is enough information? How to understand if it is enough or not?
> >
> > This one I did not get: "classifiers should be trained on different kinds of digit ground truth to assess the impact of dataset construction choices on performance". How realistic is this advice, especially for big datasets? And doesn't it suggest a condition on model training rather than on dataset construction? So I feel confused.
> >
> > I well understand that there may not be enough details available about MNIST online. If so, the authors may describe a synthetic dataset, following the guidance in the Call to Action.
> >
> > Otherwise, I am not persuaded.
> >
> > For example, in reply to W4, the authors said that the description of Imagenet is quite lengthy. But does it support the problem raised by the authors? Maybe, on the contrary, this is good because it is very verbose? Also, the authors say that the ImageNet descriptions "does not mention who the image creators were." But why and how can this information be used for training classifiers? I mean, if it is very specific, it could indeed induce some biases. But if the creators are "ordinary people", then how could it help?
> >
> > After reading the responses to W4 and W5, the position now seems like a strawman to me.
> >
> > So, I kindly ask the authors to reiterate on W4-W5.

---

### Official Review · Reviewer_4pGP · 2026-03-12

**Significance:** 3
**Argument Clarity:** 3
**Rating:** 4
**Confidence:** 3

**Questions:**

How to quantitatively evaluate the situated reliability of a ground truth, and what is the evaluation cost?

**Alternative Views Section:**

Yes

**Compliance With Llm Reviewing Policy A Conservative:**

Affirmed.

**Discussion Potential:**

3

**Final Justification:**

The rebuttal addressed my main concerns

**Paper Summary:**

This paper challenges the implicit assumption in the machine learning (ML) community that ground truth is objective and neutral, arguing instead that ground truth is a constructed product of human-technological collaboration and proposing the concept of 'situated reliability'.

**Position:**

Yes

**Position In Title:**

Yes

**Related Work:**

3

**Strengths And Weaknesses:**

Strengths:

- Systematically deconstructs four "construction sites" of ground truth: direct measurement, expert annotation, crowd-sourced consensus, and synthetic data.

- Compelling case studies including the resolution limitations of MNIST, the definition of "freshness" for Martian impact craters, and the binarization of the mushroom dataset.

- Rigorous structure with a logical progression from conceptual analysis → construction sites → collaboration mechanisms → calls to action.

Weaknesses:

- The argument that "ground truths are not neutral" has been fully discussed in many fields, with limited targeted innovation for the ML community.

- Emphasizes "reflection" and "documentation" but lacks specific technical tools or evaluation frameworks; the cost of evaluation is not discussed.

**Support:**

3

---

> ### Author Rebuttal · Authors · 2026-03-30
>
> Thank you for your thorough review of our paper.
> > Q: How to quantitatively evaluate the situated reliability of a ground truth, and what is the evaluation cost?
>
> We welcome the chance to clarify this concept. Situated reliability is not a quantitative value, but instead a key concept that helps us move from discussing reliability as a generic property to understanding it as necessarily “situated” in a particular context (see the final paragraph of Section 2).  Evaluation practices that can make the situatedness of reliability more visible include, as mentioned in the paper, evaluating against several datasets, employing a plurality of ground truths rather than upholding an illusion of one possible single ground truth, and measuring uncertainty. We also advocate for better documentation practices to characterize the relevant context (see Calls to Action 3 and 4).
>
> With regards to the evaluation cost: evaluating the performance of a model against several relevant ground truth datasets might increase cost in terms of constructing the additional datasets as well as the evaluation runtime. However, these increased costs can be weighed against the costs of inadvertently over-estimated performance and the societal and personal costs of uncritical adoption of ill-fitting ML solutions that could have potentially severe consequences (see Section 2).
>
> > W1: The argument that "ground truths are not neutral" has been fully discussed in many fields, with limited targeted innovation for the ML community.
>
> We agree that this topic has been raised (see Section 3), yet (vexingly) the current level of awareness remains insufficient, since it has not yet led to agreement on how to characterize and communicate the relevant aspects of “ground truth” data that are context-specific, situated, and (therefore) could change our conclusions if different data construction decisions were made.  It is our hope that this paper, through the recommended actions as well as dialogue (and researcher training) that views “data collection” as “data construction”, will strengthen the community’s understanding of datasets and their (in)valid uses.
>
> > W2: Emphasizes "reflection" and "documentation" but lacks specific technical tools or evaluation frameworks; the cost of evaluation is not discussed.
>
> We too are eager to see specific tools and practices that can standardize how we approach the construction and documentation of ground truth datasets.  We agree about the value of improved evaluation frameworks for context alignment to measure how well a model/dataset performs/represents the planned domain of application, but the very nature of datasets as context-specific means there will not be one single standard evaluation method that would cover all ways that reliability is situated or provide a generic reliability score.
>
> Specific steps:  We see the development of a standard vocabulary (our Call to Action 3) as the necessary first step, which can inform improved documentation methods (Call to Action 4), characterization of situated reliability (Call to Action 2), and better researcher training (Call to Action 5).  We will incorporate the discussion of evaluation cost in Call to Action 2.  We can improve the presentation of these steps to convey this envisioned sequence of solutions.

---

> > ### Author Rebuttal · Reviewer_4pGP · 2026-04-02
> >
> > My concerns have been adequately addressed.

---

### Official Review · Reviewer_rjxh · 2026-03-14

**Significance:** 4
**Argument Clarity:** 4
**Rating:** 5
**Confidence:** 4

**Questions:**

None.

**Alternative Views Section:**

Yes

**Compliance With Llm Reviewing Policy A Conservative:**

Affirmed.

**Discussion Potential:**

3

**Final Justification:**

I have updated my score based on the other reviewers.  I tend to agree with them that some of the proposed solutions appear to be somewhat abstract, but I also believe that these would be made more concrete **by the discussion prompted by the paper itself**.  So I still vote for acceptance.

**Paper Summary:**

The main position is that ground-truth data is always derived from a specific data collection and curation process, and that neglecting this simple fact can have deleterious consequences (e.g., during evaluation and application of machine learning models). The authors present ample evidence highlighting the dangers of assuming that ground-truth (data and labels alike) is universally valid, regardless of context, and suggest a number of steps to improve our community's ability to anticipate (and therefore address) this problem.

**Position:**

Yes

**Position In Title:**

Yes

**Related Work:**

4

**Strengths And Weaknesses:**

TL;DR: a well supported position paper about an important issue.

**Support/Evidence**:

- PRO: The claim is very well articulated.  I especially enjoyed reading about the wealth deleterious consequences of over-estimating the validity of "ground-truth" data scrupulously detailed in Section 3.

**Significance**:

- PRO: data is the core of machine learning and, lately, AI.  As such, this position paper is very much timely and impactful.

- CON, moderate: some of the issues listed by the authors -- I am thinking especially of (lack of) inter- and intra-annotator agreement -- are well-known in some sub-areas of ML (see for instance old works on annotator disagreement in NLP). Moreover, I think that AI companies that work on foundation models are well aware of them: data is their livelihood. E.g., the RLHF paper did focus on (and popularized) the issue of personalization/subjectivity.

- PRO: Still, I find exposure to this paper could benefit the more algorithmically minded researchers in our community, especially those that never looked at human-facing problems (say, preference elicitation) and those that earned their degree after the deep learning/scaling laws generational watershed.

**Discussion Potential**:

- PRO: this paper draws attention to a generally (although not universally) neglected issue, and as such I believe it will generate useful discussion.

- PRO: the authors propose adapting ML educational programs in Section 6 (which I believe are central); this will provide even more concrete material for discussion.

- CON: this paper is probably not very relevant for those that are already aware of this issue, e.g., researchers in hybrid decision making etc.

**Argument clarity**:

- PRO: the arguments are concise and crystal clear.

**Related works**:

- PRO: all points are supported by relevant references.

**Support:**

4

---

> ### Author Rebuttal · Authors · 2026-03-30
>
> Thank you for your careful reading of our paper.
>
> It is a good point that some of the issues we raised, such as inter-/intra-annotator agreement and ground truth subjectivity, are known in some subfields within the ML community.  Yet the current level of awareness remains insufficient, since it has not led to agreement on how to characterize and communicate the relevant aspects of “ground truth” data that are context-specific, situated, and (therefore) could change the conclusions if different data construction decisions were made. We think you refer to the RLHF (InstructGPT) paper by Ouyang et al. (2022), which showed the power of user-informed model alignment for language models.  It contains a good discussion of how the resulting fine-tuned model was influenced (constructed) by their choices about whose preferences to align to (i.e., the researchers’ own, and the 40 human contractors that were chosen to contribute feedback), and it is a good addition to the Related Work section of our paper.  Yet to date there is still no standard vocabulary (our Call to Action 3) or documentation methods (Call to Action 4) to express those limitations, nor standard practices for comparing the impact when different choices are made about who serves as the source of labels (Call to Action 2).  It is our hope that this paper, through the recommended actions as well as dialogue (and researcher training) that views “data collection” as “data construction”, will strengthen the community’s understanding of datasets and their limitations and (in)valid uses.
>
> References:
> Ouyang et al., Training language models to follow instructions with human feedback, NeurIPS, 2022. https://arxiv.org/pdf/2203.02155

---

> > ### Author Rebuttal · Reviewer_rjxh · 2026-04-04
> >
> > I will write my thoughts at a later date - though I agree with reviewer NiQW that instantiating the call for action would be useful (although in my opinion not strictly necessary). And, yes, I was referring to Ouyiang et al., apologies for not mentioning it explicitly.

---

### Decision · Program_Chairs · 2026-04-30

**Decision:**

Accept (regular)

**Comment:**

Everyone agrees on the important of the topic, generally.

One reviewer commented that those who have worked with multiple experts and dealt with inter-rater reliability know the problem well.  Certain scientific communities like sociology do this.  Another variant of this is those who have worked with disparate multi-model data.  While true, this just means some in the community are already aware of some of the issues.  Moreover, some public data sets are already well documented and "situated", so the needs for this are not as strong.

One reviewer commented that the solutions around reflection and documentation are weakly developed.  Authors acknowledged some issues and proposed revisions.  Reviewers and authors discussed ImageNet and the Wisconsin Breast Cancer Database.  The second is so old, it doesn't seem relevant.  ImageNet is a classic example of the opportunistic availability of data, where situation and documentation may not be great.

In sum, a weaker accept is supported.  Its a useful position for the community.